# NeuralFLoC: Neural Flow-Based Joint Registration and Clustering of Functional Data

**Xinyang Xiong** [* 1]  **Siyuan Jiang** [* 1]  **Pengcheng Zeng** [1]

## Abstract

Clustering functional data in the presence of phase variation is challenging, as temporal misalignment can obscure intrinsic shape differences and degrade clustering performance. Most existing approaches treat registration and clustering as separate tasks or rely on restrictive parametric assumptions. We present **NeuralFLoC**, a fully unsupervised, end-to-end deep learning framework for joint functional registration and clustering based on Neural ODE-driven diffeomorphic flows and spectral clustering. The proposed model learns smooth, invertible warping functions and cluster-specific templates simultaneously, effectively disentangling phase and amplitude variation. We establish universal approximation guarantees and asymptotic consistency for the proposed framework. Experiments on functional benchmarks show state-of-the-art performance in both registration and clustering, with robustness to missing data, irregular sampling, and noise, while maintaining scalability. Code is available at https://github.com/LastQuater/NeuralFLoC.

## 1. Introduction

Functional Data Analysis (FDA) provides a principled statistical framework for analyzing data that are naturally viewed as realizations of smooth functions or curves, such as biological signals, financial trajectories, and motion paths (Ramsay & Silverman, 2005; Ferraty & Vieu, 2006). Despite its broad applicability, FDA poses fundamental methodological challenges due to the infinite-dimensional nature of functional data, as well as practical issues such as smoothness, noise, and misalignment (Wang et al., 2016; Matuk et al., 2022). A

core challenge in addressing misalignment stems from the coexistence of two distinct sources of variability: *amplitude variation*, reflecting differences in vertical magnitude, and *phase variation*, corresponding to horizontal or temporal distortions of salient features (Srivastava et al., 2011).

### 1.1. Motivation

In this work, we focus on the challenging problem of clustering functional data in the presence of phase variation. Specifically, we consider observations of the form

$$x(t) = x_0\big(\gamma^{-1}(t)\big) + \epsilon(t),$$

where $x_0(t)$ denotes an underlying latent function (with curve-specific amplitude scaling), $\gamma(t)$ is an unknown time-warping function, and $\epsilon(t)$ represents noise. Following classical FDA formulations (Ramsay & Silverman, 2005), we assume the warping function $\gamma$ is a non-decreasing, continuous map on a compact domain $\mathcal{D} = [a, b]$ with boundary conditions $\gamma(a) = a$ and $\gamma(b) = b$. Without loss of generality, we take $\mathcal{D} = [0, 1]$.

Clustering such data is essential for uncovering latent functional patterns and heterogeneity. However, substantial phase variation can obscure the intrinsic shape differences between curves, causing conventional clustering algorithms to group functions according to misalignment rather than underlying structure, thereby leading to degraded performance (Marron et al., 2015).

Existing approaches to this problem largely fall into two categories. The first strategy treats registration (alignment) and clustering as two separate steps (Sangalli et al., 2010). A typical pipeline first estimates warping functions $\gamma(t)$'s—often by aligning curves to a global template using Dynamic Time Warping (DTW) (Sakoe & Chiba, 1978) or the Square Root Velocity Function (SRVF) framework (Srivastava et al., 2011)—and then applies standard clustering algorithms such as $K$-means (Tucker et al., 2013) on the aligned $x(\gamma(t))$'s. While intuitive, this two-stage approach suffers from several drawbacks: (1) the alignment step depends on a prespecified or data-dependent template, which may introduce bias; (2) registration errors propagate to the clustering stage; (3) many elastic registration methods are computationally intensive or fail to guarantee monotonicity and invertibility

---
[*]Equal contribution [1]ShanghaiTech University, Institution of Mathematical Sciences, Shanghai, China. Correspondence to: Pengcheng Zeng <zengpch@shanghaitech.edu.cn>.

*Proceedings of the 43rd International Conference on Machine Learning*, Seoul, South Korea. PMLR 306, 2026. Copyright 2026 by the author(s).

of the warping functions, properties that are essential for preserving the physical topology of time (Vercauteren et al., 2009).

The second strategy seeks to jointly perform registration and clustering within a unified statistical framework (Liu & Yang, 2009; Wu & Hitchcock, 2016; Zeng et al., 2019; Gaffney & Smyth, 2004). Despite their conceptual appeal, existing joint models exhibit notable limitations: (1) they often impose restrictive assumptions on the warping functions, such as simple time shifts (Liu & Yang, 2009), Dirichlet process priors (Wu & Hitchcock, 2016), or parametric mixed-effects structures (Zeng et al., 2019); (2) they are difficult to extend to multivariate or vector-valued functional data; (3) they rely on computationally expensive inference procedures, limiting scalability to large datasets.

To address these challenges, we propose **NeuralFLoC**, a deep learning framework for joint diffeomorphic registration and spectral clustering of functional data using neural flows. The method ensures smooth, invertible transformations without explicit distributional assumptions, scales efficiently, and naturally extends to multivariate settings. To our knowledge, this is the first end-to-end deep learning approach to simultaneous functional registration and clustering.

### 1.2. Related Work

**Curve Registration and the SRVF Framework.** Curve registration aims to remove phase variability by estimating optimal warping functions. The Square Root Velocity Function (SRVF) framework (Srivastava et al., 2011) is a landmark contribution, as it transforms the Fisher–Rao Riemannian metric into a standard $\mathbb{L}^2$ metric, enabling efficient elastic shape analysis. Nevertheless, traditional SRVF-based methods typically rely on dynamic programming or iterative optimization, which can be computationally demanding (Srivastava & Klassen, 2016). More recently, neural network-based approaches have been explored (Chen & Srivastava, 2021), but these are often employed as preprocessing steps and remain decoupled from downstream learning objectives.

**Neural Ordinary Differential Equations.** Neural Ordinary Differential Equations (Neural ODEs) model hidden representations as continuous-time flows parameterized by neural networks (Chen et al., 2018). They have demonstrated strong performance in time-series modeling and generative learning (Rubanova et al., 2019). In this work, we leverage Neural ODEs to construct smooth, invertible, and biologically plausible time-warping transformations for functional data registration, drawing connections to continuous normalizing flows and diffeomorphic mappings (Dupont et al., 2019).

**Functional Clustering.** Functional clustering seeks to group curves according to shared structural characteristics. Classical approaches cluster basis coefficients obtained from spline or wavelet expansions (Ramsay & Silverman, 2005). More recent developments incorporate deep representation learning, such as Deep Embedding Clustering (DEC) (Xie et al., 2016). However, most existing methods implicitly assume that functional observations are already aligned or focus exclusively on amplitude variation, limiting their effectiveness when pronounced phase variability is present (Marron et al., 2015).

### 1.3. Contributions

Our main contributions are as follows:

- *A continuous diffeomorphic registration model* based on Neural ODEs, which guarantees smooth, invertible warping functions by construction.
- *A fully unsupervised deep learning framework that jointly aligns functions and infers clusters* by integrating SRVF-based registration with spectral clustering, separating phase and amplitude variation through cluster-specific templates.
- *Theoretical guarantees*—including universal approximation and consistency—and *a scalable optimization procedure* with linear-time sample complexity and constant memory usage, empirically validated on diverse functional datasets.

## 2. Preliminaries

**Warping Space and Diffeomorphisms.** To represent nonlinear temporal misalignments, we consider the space of warping functions $\Gamma$. Following (Ramsay & Silverman, 2005), a valid warping function must preserve the temporal topology. We usually define the warping space $\Gamma$ as the set of all positive boundary-preserving diffeomorphisms on the unit interval $[0, 1]$:

$$\Gamma = \{\gamma : [0, 1] \to [0, 1] \mid \gamma(0) = 0, \gamma(1) = 1, \dot{\gamma} > 0\}. \tag{1}$$

The condition $\dot{\gamma} > 0$ ensures that $\gamma$ is strictly monotonic and thus invertible, which is a fundamental requirement for meaningful time-warping in FDA (Marron et al., 2015).

**Neural ODEs.** A Neural ODE models the evolution of a latent state $\mathbf{h}(t)$ as a continuous-time dynamical system (Chen et al., 2018). The dynamics are governed by a vector field parameterized by a neural network $f(\mathbf{h}(t), t, \theta)$:

$$\frac{d\mathbf{h}(t)}{dt} = f(\mathbf{h}(t), t, \theta), \quad \mathbf{h}(0) = \mathbf{h}_0. \tag{2}$$

The solution at time $t$ can be obtained via an ODE solver: $\mathbf{h}(t) = \mathbf{h}_0 + \int_0^t f(\mathbf{h}(s), s, \theta)ds$. In our framework, we

leverage the continuous flow of Neural ODEs to parameterize the warping function $\gamma(t)$, ensuring smoothness and strict monotonicity through the choice of the vector field $f$ (Rubanova et al., 2019).

**SRVF.** To achieve alignment that is invariant to reparameterization, we adopt the SRVF framework (Srivastava et al., 2011). For a smooth function $x : [0, 1] \to \mathbb{R}$, its SRVF is defined as:

$$q(t) = \text{sign}(\dot{x}(t))\sqrt{|\dot{x}(t)|}. \tag{3}$$

The primary advantage of SRVF is that the Fisher-Rao Riemannian metric on the space of functions is transformed into the standard $\mathbb{L}^2$ metric in the SRVF space (Srivastava & Klassen, 2016). Specifically, for any warping function $\gamma \in \Gamma$, the SRVF of the warped function $x(\gamma(t))$ is given by $(q \circ \gamma)(t)\sqrt{\dot{\gamma}(t)}$. This isometry allows us to compute the registration loss using the standard Euclidean distance in the SRVF space while maintaining its geometric significance.

## 3. Methodology

Consider functional observations $\{x_i(t)\}_{i=1}^N$, each sampled at $T$ time points, $\mathbf{x}_i = (x_i(t_{i1}), \ldots, x_i(t_{iT}))$. Such data often exhibit significant phase variation (Figure 2(a)), which can obscure shape differences and hinder direct clustering (Ramsay & Silverman, 2005). Our goal is to jointly recover distinct shape patterns and nonlinear time warpings $\gamma_i \in \Gamma$ that align similar curves. To this end, we estimate optimal warping functions $\{\gamma_i\}$—parameterized via a Neural ODE—and cluster assignments by minimizing a synergistic objective. This joint formulation allows registration to be guided by cluster-specific templates, while clustering operates on phase-corrected representations (Marron et al., 2015).

The proposed framework, **NeuralFLoC**, consists of three integrated modules: a 1D-CNN encoder for feature extraction, a Neural ODE block for continuous-time diffeomorphic warping, and a Fourier-based soft-assignment clustering layer (Figure 1).

### 3.1. Latent Initialization via 1D-CNN

1D convolutional neural networks (1D-CNNs) effectively capture local morphological patterns in functional data (Kiranyaz et al., 2015; Jiang et al., 2025). Our encoder, parameterized by $\Theta_{\text{enc}}$, defines a nonlinear mapping $\Phi : \mathcal{X} \to \mathcal{Z}$ that projects a raw sequence $\mathbf{x}_i \in \mathbb{R}^{1 \times T}$ into a latent space $\mathcal{Z} \in \mathbb{R}^C$, where $C$ denotes the number of clusters (see section 6 on how to determine $C$). The latent vector $\mathbf{z}_i$ is computed as:

$$\mathbf{z}_i = \text{ReLU}(W \, \text{Enc}(\mathbf{x}_i; \Theta_{\text{enc}}) + \mathbf{b}), \tag{4}$$

where $\text{Enc}(\cdot)$ denotes the 1D-CNN layers, and $W, \mathbf{b}$ are parameters of a final fully-connected layer. This $\mathbf{z}_i$ serves as the initial condition for the subsequent dynamics, ensuring that alignment begins from a shape-aware representation.

### 3.2. Continuous Time-Warping via Neural ODEs

The core alignment is performed by a Neural ODE warper that models time-warping as a continuous flow $\tau_i(t)$ governed by a learnable velocity field $f$. To guarantee diffeomorphic warps in $\Gamma$, we solve the initial value problem (IVP) (Chen et al., 2018):

$$\frac{d\tau_{ij}(t)}{dt} = f(\tau_{ij}(t), t, z_{ij}; \Theta_{\text{ode}}), \quad \tau_{ij}(0) = z_{ij}, \tag{5}$$

where $f$ is a multi-layer perceptron, and $z_{ij}$ is the $j$-th element of $\mathbf{z}_i$. A Softplus activation is applied to the output of $f$ to enforce strict monotonicity ($\dot{\gamma} > 0$), as required for valid warping (Srivastava & Klassen, 2016). The normalized warping function is obtained via boundary scaling:

$$\hat{\gamma}_{ij}(t) = \frac{\tau_{ij}(t) - \tau_{ij}(0)}{\tau_{ij}(1) - \tau_{ij}(0)}. \tag{6}$$

To handle multiple shape modes, the instance-specific warping is defined as a mixture of class-specific flows weighted by assignment probabilities $p_{ij}$:

$$\gamma_i(t) = \sum_{j=1}^{C} p_{ij} \, \hat{\gamma}_{ij}(t). \tag{7}$$

Initial assignments $\{\mathbf{p}_i\} = \{p_{i1}, ..., p_{iC}\}$ are set uniformly. Note that the vector $\mathbf{z}_i \in R^C$ (Eq. 4) is not a free latent embedding: its $j$-th coordinate initializes the $j$-th cluster-specific ODE flow whose mixture forms the final warp (Eq. 7). Thus, the dimensionality is structural rather than a representational bottleneck: the 1D-CNN encoder itself has full expressive power, while the final linear layer simply reorganizes its outputs into the flow-related C-dimensional structure. The aligned function $\tilde{x}_i(t) = x_i(\gamma_i(t))$ is computed using 1D linear interpolation, which ensures stable gradient flow during backpropagation. The complete procedure is outlined in Algorithm 1.

### 3.3. Clustering Assignment in Spectral Domain

We perform clustering in a low-dimensional spectral domain by projecting aligned curves onto a Fourier basis and applying a differentiable soft-assignment mechanism.

**Fourier Representation.** The aligned data $\tilde{x}_i(t)$ is approximated using a Fourier basis expansion $\{\phi_k(t)\}_{k=1}^K$:

$$\tilde{x}_i(t) \approx \sum_{k=1}^{K} a_{ik}\phi_k(t), \tag{8}$$

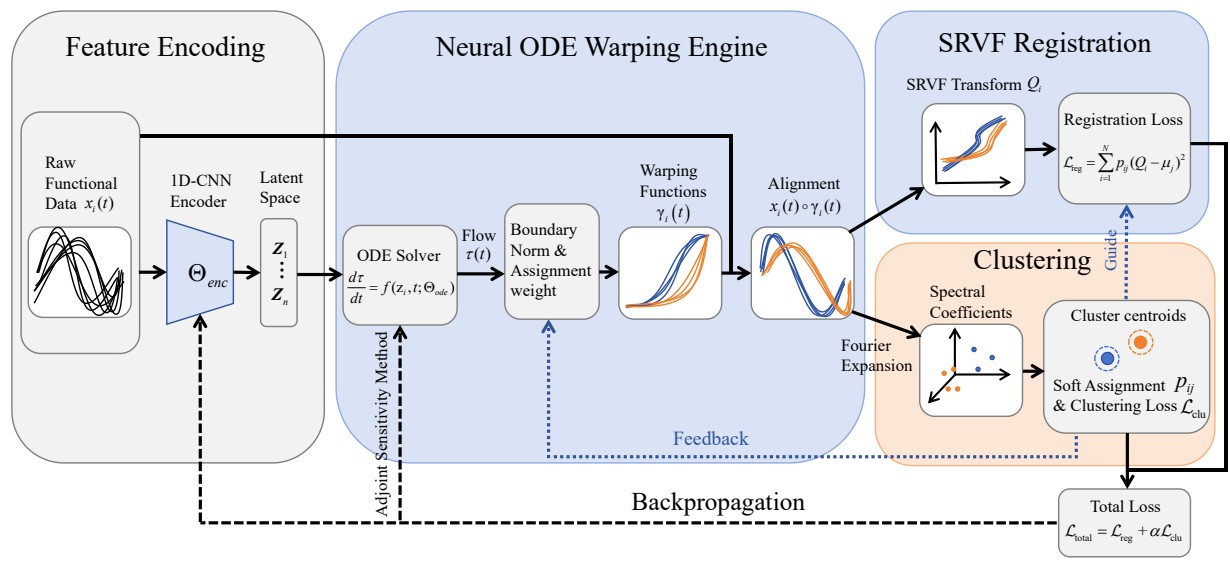

*Figure 1.* Overview of **NeuralFLoC**. A 1D-CNN encodes raw functional data. A Neural ODE generates smooth, invertible warping functions $\gamma_i(t)$ to align curves, and the aligned features are clustered via a Fourier-based soft-assignment module. The model is trained end-to-end for joint registration and clustering.

---

**Algorithm 1** Time-warping Module

---

1: **Input:** Latent vectors $\{\mathbf{z}_i\}$, ODE parameters $\Theta_{\text{ode}}$, Assignments $\{\mathbf{p}_i\}$.
2: Solve IVP using Adjoint Method: $\tau_i(t) = \text{ODESolver}(\mathbf{z}_i, t, \Theta_{\text{ode}})$.
3: Compute boundary-normalized warping $\hat{\gamma}_i(t)$ via Eq. 6.
4: Compute instance-specific warping $\gamma_i(t) = \sum p_{ij}\hat{\gamma}_{ij}(t)$.
5: Apply $\gamma_i(t)$ to $x_i(t)$ via linear interpolation to get $\tilde{x}_i(t)$.
6: **Output:** Warping functions $\{\gamma_i\}$, Aligned curves $\{\tilde{x}_i\}$.

---

where $\mathbf{a}_i \in \mathbb{R}^K$ are the expansion coefficients (default $K = 10$). This projection filters high-frequency noise while preserving essential shape features (Ramsay & Silverman, 2005) and functions analogously to spectral embedding in traditional spectral clustering.

**Soft Assignment.** Similarity between coefficient vector $\mathbf{a}_i$ and cluster centroids $\{\mathbf{c}_j\}_{j=1}^C$ is measured via a Student's $t$-distribution kernel (Xie et al., 2016):

$$p_{ij} = \frac{(1 + \|\mathbf{a}_i - \mathbf{c}_j\|^2/\epsilon_0)^{-\frac{\epsilon_0+1}{2}}}{\sum_{j'}(1 + \|\mathbf{a}_i - \mathbf{c}_{j'}\|^2/\epsilon_0)^{-\frac{\epsilon_0+1}{2}}}, \quad (9)$$

with $\epsilon_0 = 1$. The resulting soft-assignment probability $p_{ij}$ is fed back into the warping module (Eq. 7), enabling the model to refine alignments using current cluster assignments. Note that the cluster centroids $\mathbf{C}$ are also learnable parameters, initialized via K-means on the Fourier coefficients of

the raw curves and subsequently optimized via the joint loss (Eq. 12).

### 3.4. Objective Function Formulation

We propose a unified objective that jointly learns diffeomorphic temporal alignments and cluster assignments, enabling class-specific functional registration while preserving discriminative shape information.

**Cluster-Conditional SRVF Registration.** To measure alignment quality under a reparameterization-invariant geometry, we adopt the square-root velocity function (SRVF) representation (Srivastava et al., 2011). For a warped curve $\tilde{x}_i(t) = x_i(\gamma_i(t))$, its SRVF is

$$Q_i(t) = \text{sign}(\dot{\tilde{x}}_i(t))\sqrt{|\dot{\tilde{x}}_i(t)|}.$$

Instead of aligning all curves to a global template, we introduce cluster-conditional alignment targets. For each cluster $j$, we define a weighted Karcher mean in SRVF space,

$$\mu_j = \frac{\sum_{k=1}^N p_{kj} Q_k}{\sum_{k=1}^N p_{kj}},$$

and formulate the registration loss as the expected within-cluster dispersion,

$$\mathcal{L}_{\text{reg}} = \sum_{i=1}^N \sum_{j=1}^C p_{ij} \|Q_i - \mu_j\|_2^2. \quad (10)$$

This encourages class-aware alignment and effectively disentangles phase and amplitude variation.

**Algorithm 2** Joint Registration and Clustering Training

1: **Input:** Data $\{x_i\}$, Cluster number $C$, Weight $\alpha$.
2: **Initialize:** $\Theta_{\text{enc}}, \Theta_{\text{ode}}$, and centroids **C** via K-means on raw Fourier coefficients.
3: **while** not converged **do**
4:    Calculate latent features $\mathbf{z}_i$ via Eq. 4.
5:    $\{\gamma_i\}, \{\tilde{x}_i\}$ = Time-warping($\{\mathbf{z}_i\}, \Theta_{\text{ode}}, \{\mathbf{p}_i\}$).
6:    Update soft assignments $p_{ij}$ and target $q_{ij}$.
7:    Compute $\mathcal{L}_{\text{total}} = \mathcal{L}_{\text{reg}} + \alpha\mathcal{L}_{\text{clu}}$.
8:    Update parameters using Adam and Adjoint Method gradients.
9: **end while**
10: **Output:** Optimal warping functions $\{\gamma_i\}$ and assignments $\mathbf{p}_i$.

**Self-Paced Clustering.** Cluster assignments are learned via soft probabilities $p_{ij}$ and refined using a self-training objective inspired by deep embedded clustering (DEC) (Xie et al., 2016). The target distribution

$$q_{ij} = \frac{p_{ij}^2/g_j}{\sum_{j'} p_{ij'}^2/g_{j'}}, \qquad g_j = \sum_i p_{ij},$$

emphasizes confident assignments and avoids degenerate solutions. The clustering loss is defined as

$$\mathcal{L}_{\text{clu}} = \text{KL}(Q\|P) = \sum_{i=1}^{N}\sum_{j=1}^{C} q_{ij}\log\frac{q_{ij}}{p_{ij}}. \qquad (11)$$

**Joint Objective.** The final end-to-end objective is

$$\mathcal{L}_{\text{total}} = \mathcal{L}_{\text{reg}} + \alpha\,\mathcal{L}_{\text{clu}}, \qquad (12)$$

where $\alpha$ balances geometric alignment and cluster discrimination (default $\alpha = 0.01$). The two terms are optimized jointly, allowing clustering to guide class-specific registration while improved alignment yields shape-pure representations for more accurate clustering.

### 3.5. Optimization, Complexity, and Extension

The model parameters consist of three groups: encoder weights $\Theta_{\text{enc}}$, ODE parameters $\Theta_{\text{ode}}$, and cluster centroids **C**. Gradients of the loss w.r.t. $\Theta_{\text{ode}}$ are computed via the adjoint sensitivity method (Chen et al., 2018), enabling backpropagation through the ODE solver with $O(1)$ memory overhead. The full training loop is outlined in Algorithm 2.

**Complexity analysis.** The framework achieves efficiency through three modular steps: (1) 1D-CNN encoding runs in $O(N \cdot T)$; (2) Neural ODE warping requires $T$ function evaluations while maintaining $O(1)$ memory via the adjoint method; (3) joint optimization computes the SRVF registration loss in $O(N \cdot C \cdot T)$ and the spectral clustering loss

in $O(N \cdot C \cdot K)$, where $C$ is the number of clusters and $K$ the Fourier basis size. The total per-iteration complexity is $O\big(N(CK+CT+T)\big)$, scaling linearly with sample size $N$. This represents a substantial improvement over traditional $O(T^2)$ dynamic programming methods (Sakoe & Chiba, 1978), particularly when $T \gg N$.

**Extension to multivariate functional data.** Our framework naturally handles $d$-variate inputs ($d \geq 2$) by taking $\mathbf{x}_i \in \mathbb{R}^{d\times T}$ and assuming a shared warping process across dimensions, without introducing additional structural assumptions.

## 4. Theoretical Analysis

We next establish the theoretical foundation of our joint framework by presenting two theorems on the approximation consistency of the neural registration model and the consistency of the joint registration-clustering estimator. Detailed proofs are provided in Appendix A.

**Theorem 4.1** (Approximation Consistency of Neural Registration). *Let $\Gamma$ denote the space of boundary-preserving diffeomorphisms on $[0, 1]$. Assume that:*

*(A1) The encoder class $\{\Phi(\cdot; \Theta_{\text{enc}})\}$ (1D-CNNs) is dense in $C(\mathcal{X}, \mathbb{R}^d)$ on compact subsets of the input space $\mathcal{X}$.*

*(A2) The Neural ODE vector field class $\{f(\cdot, \cdot, \cdot; \Theta_{\text{ode}})\}$ is dense in the space of continuous and uniformly Lipschitz functions on $[0, 1] \times [0, 1] \times \mathbb{R}^d$.*

*(A3) The ground-truth warping functions $\gamma_i^*$ are generated by an ODE with a Lipschitz continuous velocity field and satisfy $\gamma_i^* \in \Gamma$.*

*(A4) Each observed curve $x_i$ is absolutely continuous with bounded derivative, so that its SRVF $q_i$ exists and is continuous.*

*Then, for any $\varepsilon > 0$, there exist parameters $(\Theta_{\text{enc}}, \Theta_{\text{ode}})$ such that the induced warping functions $\hat{\gamma}_i$ satisfy*

$$\big|R(\gamma^*) - R(\hat{\gamma})\big| < \varepsilon,$$

*where $R(\gamma)$ denotes the total SRVF-based registration loss.*

**Remark 1.** Assumptions (A1)–(A2) are grounded in classical universal approximation theorems for convolutional networks and Neural ODEs (Hornik et al., 1989; Teshima et al., 2020) on compact domains. Assumptions (A3)–(A4) mirror standard premises in elastic functional data analysis, where smooth diffeomorphic warpings capture phase variability and observations exhibit piecewise smoothness (Ramsay & Silverman, 2005; Srivastava et al., 2011). These conditions are empirically verifiable through learned warping monotonicity and ODE stability checks. Under these

mild assumptions, Theorem 4.1 provides a theoretical foundation: with adequate network capacity, the Neural ODE registration module can approximate any admissible warping arbitrarily well in SRVF metric. This establishes a principled deep-learning alternative to classical elastic registration, aligning with the empirical robustness reported in Section 6.

**Theorem 4.2** (Consistency of Joint Registration and Clustering). *Let $\{x_i\}_{i=1}^N$ be functional observations generated from a finite mixture of $C$ latent clusters. Assume that:*

(B1) (**Cluster Separability**) *After perfect registration, the population Fourier coefficient vectors $\mathbf{a}_i^* \in \mathbb{R}^K$ form $C$ well-separated clusters, i.e.,*

$$\min_{j \neq j'} \|\mu_j^* - \mu_{j'}^*\|_2 \geq \Delta > 0,$$

*where $\mu_j^*$ denotes the population centroid of cluster $j$.*

(B2) (**Noise Model**) *The observational noise is i.i.d. Gaussian with finite second moments.*

(B3) (**Identifiability**) *The joint population risk admits a unique minimizer (up to label permutation).*

*Then, as $N \to \infty$, the estimated soft assignments $\hat{p}_{ij}$ converge in probability to the true assignments $p_{ij}^*$, and the empirical joint objective $\mathcal{L}_{\text{total}}$ converges to its population minimum.*

**Remark 2.** Assumption (B1) posits cluster separability after phase removal—a standard condition in clustering theory that holds when phase variability dominates shape differences, as in biomedical signals and motion trajectories. Assumptions (B2)–(B3) specify a conventional noise model and joint objective identifiability, both assessable through residual diagnostics and partition stability across initializations. Theorem 4.2 shows that under these conditions, joint registration and clustering is statistically well-posed: increasing sample size causes alignment and clustering to mutually reinforce, yielding stable partitions under noise and severe phase variability. This explains the method's robustness to data imperfections and validates its use in large-scale unsupervised functional data analysis where both tasks are essential (Section 6).

## 5. Experiments

### 5.1. Experimental Settings

**Datasets.** We evaluate our method on diverse datasets from the UCR Time Series Classification Archive (Dau et al., 2019), selected for their phase variation and use in functional data benchmarks (Jiang et al., 2025). These datasets span multiple domains and include both multi-class and multivariate configurations:

- **Shapes** (image-derived): 1D sequences (1095 samples, 1024 points) converted from 2D shape boundaries, evaluated on a multi-class problem focusing on classes 4–5.
- **Wave** (sensor/motion): Accelerometer gesture recordings analyzed in both univariate ($d = 1$, X-axis) and multivariate ($d = 3$, X-, Y-, and Z-axis) configurations (1120 samples, 315 points), using a binary classification setup (classes 2 and 8).
- **Symbols** (handwriting): Pen trajectory data evaluated on both binary (2-class) and multi-class (3-class) subsets with 398 time points, containing 343 and 510 samples respectively.

**Evaluation Metrics.** We assess registration quality using the Adjusted Total Variance (ATV) (Jiang et al., 2025), which quantifies compactness within aligned classes relative to their separation—lower ATV indicates better reduction of phase variability. For clustering, we report Accuracy (ACC) and Normalized Mutual Information (NMI) (Strehl & Ghosh, 2002), where higher values reflect better alignment between predicted clusters and true labels. Formal definitions are provided in Appendix B.

**Baselines.** We compare our proposed framework against traditional FDA techniques and recent deep learning models:

- **Traditional FDA**: Functional principal component analysis (FPCA) and B-spline expansion (Ramsay & Silverman, 2005), both followed by $k$-means.
- **Unsupervised Registration**: SrvfRegNet (Chen & Srivastava, 2021), a CNN-based model that aligns raw functional curves.
- **Deep Clustering**: Functional autoencoder (FAE) (Wu et al., 2024) and RandomNet (Li et al., 2024), a state-of-the-art deep clustering model for functional data.
- **Graph-based**: K-graph (Boniol et al., 2025), which performs clustering via graph-based manifold learning.
- **Discrete Baseline**: We replace our Neural ODE warping module with SrvfRegNet to compare continuous-flow warping against a discrete registration network.
- **Joint Methods**: JPCCA(Gaffney & Smyth, 2004), BSRC(Wu & Hitchcock, 2016) and SRC(Zeng et al., 2019).These are classical staistical methods that perform simultaneous functional registration and clustering.

For a fair comparison, all sequential registration-then-clustering pipelines are evaluated by combining SrvfRegNet with each downstream clustering method (FPCA, B-spline, FAE, RandomNet, K-graph). The variant of our method that uses SrvfRegNet instead of the Neural ODE warper is denoted as *Ours (N-ODE → SrvfRegNet)*, and the variant that uses short-time Fourier transform (STFT) features instead of the Fourier basis is denoted as *Ours (Fourier → STFT)*.

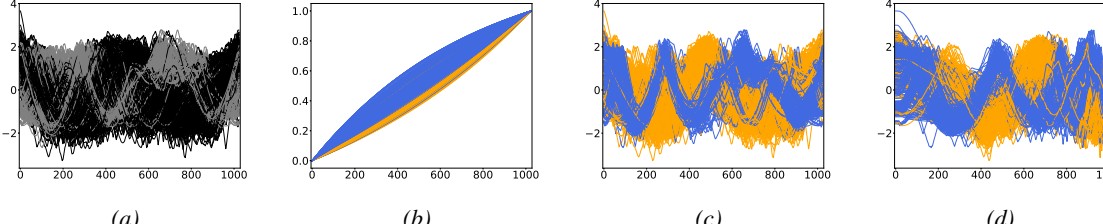

*Figure 2.* (a) Raw data in **Shapes** dataset with two ground-truth groups indicated by black and grey. (b) Learned warping functions $\gamma(t)$'s by *Ours*. (c) Alignment by *Ours*. (d) Alignment by *Ours (N-ODE → SrvfRegNet)*. Colored curves in (b)-(d) indicate the clustering results.

**Implementation Details.** Our feature encoder uses three 1D-CNN blocks (channels: $16 \rightarrow 32 \rightarrow 64$, kernel size 3). The Neural ODE velocity field $f$ is an MLP with architecture $(C + 1) \rightarrow 64 \rightarrow 64 \rightarrow C$, where $C$ equals the number of clusters (set to the ground-truth value). We employ ELU activations (Clevert et al., 2016) to maintain smooth gradients. For spectral clustering we fix $K = 10$ Fourier basis functions and use a constant loss weight $\alpha = 0.01$.

Our model is optimized with Adam (Kingma & Ba, 2015) (initial learning rate $10^{-3}$), decayed by a factor of 10 every 100 epochs for a total of 300 epochs. Each experiment is repeated 10 times; results report mean. Hyperparameters of baseline methods follow their original papers or default settings. The number of clusters $C$ is set to the true number of classes for all methods, ensuring a fair comparison.

### 5.2. Experimental Results

**Performance Visualization.** As shown in Figure 2 (using the **Shapes** dataset), our method successfully disentangles phase and amplitude variation—particularly for curves in the blue cluster. In the raw data (Figure 2(a)), nonlinear temporal shifts obscure ground-truth class boundaries. The warping functions learned by our method (Figure 2(b)) satisfy the required monotonicity and diffeomorphism constraints, demonstrating the flexibility of the ODE-based velocity field. After joint alignment and spectral clustering (Figure 2(c)), curves within each cluster are synchronized to a common template, substantially reducing intra-cluster total variation while increasing inter-cluster separation, which directly improves clustering performance. Performance visualization for the remaining datasets are provided in Figures A.5 - A.8 in Appendix C.

**Performance Comparison.** Table 1 summarizes the registration and clustering results across all datasets. In contrast to sequential pipelines (e.g., SrvfRegNet followed by clustering) that risk "over-warping" and do not use cluster structure to inform alignment, our joint framework (*Ours*) yields superior registration and higher clustering accuracy. To verify whether these benefits originate from our continuous-flow formulation or merely from joint optimiza-

tion, we compare our full Neural ODE model with three types of alternatives: a discrete-warping variant that substitutes the ODE with SrvfRegNet but retains joint training (*Ours (N-ODE → SrvfRegNet)*); several representative joint registration–clustering methods (BSRC, JPCCA, SRC); and a feature-engineering variant that uses short-time Fourier transform features instead of the Fourier basis (*Ours (Fourier → STFT)*). Quantitatively, the complete model outperforms the discrete variant on every dataset, with the most pronounced gap on Wave (d=1). On this dataset, which exhibits minimal phase variability, the discrete variant severely underperforms because jointly training SrvfRegNet without the ODE's smoothness constraints causes optimization instability, weaker regularization that encourages over-warping even when warps are nearly trivial, and difficulty in adapting pretrained alignment alongside clustering. Our continuous Neural ODE warper avoids these issues entirely, achieving the highest accuracy, and this case therefore clearly highlights the superiority of continuous flows in maintaining stable alignment even when the true warping structure is simple. The classical joint baselines, despite also solving registration and clustering simultaneously, all yield markedly lower clustering accuracy. This demonstrates that the observed gains do not arise from the joint paradigm itself, but from our continuous diffeomorphic warping, mixture-of-flows formulation, and end-to-end optimization. Furthermore, replacing the Fourier basis with STFT features consistently degrades performance. After SRVF-based phase alignment, most residual amplitude variation resides in a low-frequency subspace; STFT overemphasizes high-frequency artifacts from imperfect warping, whereas the truncated Fourier basis provides a natural geometry-consistent denoising effect. Qualitatively, Figure 2(c,d) on the **Shapes** dataset shows that our method produces smoother warpings and clearer cluster separation, visually justifying the improved metrics. These findings highlight the benefit of continuous-flow warping in the joint framework.

**Ablation Study.** The bottom rows of Table 1 ablate our method's core components: Neural ODE-based warping with SRVF transformation, and Fourier spectral clustering with soft assignment. Replacing the ODE warper with dis-

*Table 1.* Quantitative comparison of registration and clustering performance. **Bold** numbers indicate best results.

| Methods | Shapes | | | Wave ($d=1$) | | | Wave ($d=3$) | | | Symbols (2 classes) | | | Symbols (3 classes) | | |
|---|---|---|---|---|---|---|---|---|---|---|---|---|---|---|---|
| | ATV | ACC | NMI | ATV | ACC | NMI | ATV | ACC | NMI | ATV | ACC | NMI | ATV | ACC | NMI |
| SrvfRegNet + FPCA | 23.2 | 0.722 | 0.180 | 4.8 | 0.990 | 0.922 | 13.5 | 0.970 | 0.825 | 7.7 | 0.760 | 0.224 | 2.4 | 0.805 | 0.602 |
| SrvfRegNet + BSpline | 23.2 | 0.697 | 0.201 | 4.8 | 0.988 | 0.909 | 13.5 | 0.744 | 0.208 | 7.7 | 0.630 | 0.049 | 2.4 | 0.761 | 0.551 |
| SrvfRegNet + FAE | 23.2 | 0.521 | 0.000 | 4.8 | 0.511 | 0.001 | 13.5 | 0.512 | 0.000 | 7.7 | 0.521 | 0.002 | 2.4 | 0.362 | 0.003 |
| SrvfRegNet + RandomNet | 23.2 | 0.723 | 0.182 | 4.8 | 0.991 | 0.921 | 13.5 | 0.971 | 0.825 | 7.7 | 0.760 | 0.226 | 2.4 | 0.805 | 0.602 |
| SrvfRegNet + K-Graph | 23.2 | 0.918 | 0.632 | 4.8 | 0.950 | 0.740 | 13.5 | 0.947 | 0.714 | 7.7 | 0.524 | 0.002 | 2.4 | 0.656 | 0.553 |
| SRC | 18.5 | 0.807 | 0.284 | 10.5 | 0.884 | 0.506 | 21.6 | 0.573 | 0.099 | 7.3 | 0.581 | 0.031 | 5.5 | 0.508 | 0.200 |
| BSRC | 18.5 | 0.755 | 0.279 | 6.1 | 0.852 | 0.443 | 28.5 | 0.742 | 0.244 | 19.8 | 0.647 | 0.064 | 5.7 | 0.831 | 0.594 |
| JPCCA | 12.5 | 0.622 | 0.000 | 18.1 | 0.503 | 0.005 | 46.7 | 0.531 | 0.053 | 11.4 | 0.892 | 0.518 | 10.3 | 0.355 | 0.000 |
| *Ours (Fourier→STFT)* | 15.1 | 0.822 | 0.398 | 5.4 | 0.988 | 0.910 | 10.9 | 0.965 | 0.783 | 5.8 | 0.886 | 0.520 | 2.6 | 0.849 | 0.671 |
| *Ours (N-ODE → SrvfRegNet)* | 8.6 | 0.908 | 0.599 | 7.6 | 0.552 | 0.009 | 14.0 | 0.796 | 0.481 | **2.9** | 0.873 | 0.495 | 1.7 | 0.910 | 0.779 |
| *Ours* | **8.1** | **0.937** | **0.651** | **4.1** | **0.993** | **0.941** | **8.5** | **0.992** | **0.942** | 3.1 | **0.930** | **0.639** | **1.6** | **0.951** | **0.835** |
| *Ours w/o Reg* | - | 0.884 | 0.471 | - | 0.948 | 0.744 | - | 0.954 | 0.753 | - | 0.787 | 0.253 | - | 0.849 | 0.652 |
| *Ours w/o Clu* | 24.1 | - | - | 5.8 | - | - | 14.0 | - | - | 7.0 | - | - | 2.9 | - | - |

crete SrvfRegNet (*Ours (N-ODE → SrvfRegNet)*) causes a consistent performance drop, confirming the advantage of continuous-flow registration. Removing registration entirely (*Ours w/o Reg*) severely degrades clustering, while disabling clustering (*Ours w/o Clu*) impairs alignment. Replacing the Fourier basis with short-time Fourier transform features (*Ours (Fourier→STFT)*) also leads to worse performance. As shown in Table A.2 in Appendix C, paired $t$-tests over 10 random seeds confirm that ablating any component yields a statistically significant decline ($p < 0.01$ for most comparisons). These results demonstrate that both registration and clustering modules are essential to our framework's performance.

**Sensitivity Analysis.** We analyze **NeuralFLoC**'s sensitivity to the loss weight $\alpha$ and the number of Fourier basis functions $K$ in Figure 3. Increasing $\alpha$ yields modest gains up to $\alpha \approx 0.010$, after which performance saturates (left panel). Larger $K$ improves results until $K \approx 10$; performance remains stable for $K \leq 14$ and degrades thereafter (right panel). All reported experiments use $\alpha = 0.010$ and $K = 10$.

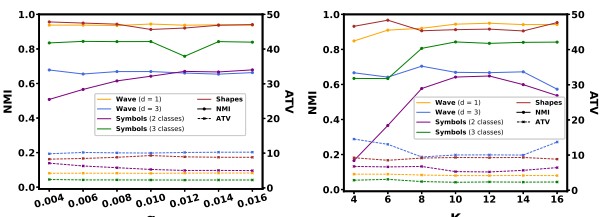

*Figure 3.* **Left**: Sensitivity to loss weight $\alpha$. **Right**: Sensitivity to number of basis functions $K$

## 6. Discussion

**Handling Unknown Cluster Number in Practice.** Although for fair comparison we used the ground-truth cluster number $C$ across all baselines, our method can natu-

rally select $C$ in practice via an elbow analysis on the total variation (TV) metric from Fisher–Rao alignment (Chen & Srivastava, 2021). The rationale is that alignment quality, as measured by TV, is optimized when the cluster number is correctly chosen. Importantly, TV does not rely on ground-truth labels, making it a principled, method-specific criterion for cluster selection. Applying this elbow analysis to our five datasets yields cluster number estimates of $(3, 2, 2, 2, 3)$, closely matching the true cluster numbers $(2, 2, 2, 2, 3)$. This demonstrates that our framework offers a data-driven approach for choosing $C$ when it is unknown.

**Robustness to Missing Data.** We evaluate robustness to missing observations by randomly removing $5\%$, $10\%$, and $20\%$ of the points from half of each dataset, imputing the missing values with Fourier splines (Wahba, 1990). Using the same hyperparameter configuration, our model maintains registration and clustering performance comparable to that obtained on the complete data (left panel, Figure 4).

**Robustness to Irregular Sampling.** We simulate irregular sampling by applying temporal perturbations to all datasets. First, each raw curve is fitted with a Fourier spline (Wahba, 1990); then, an uneven grid is generated using the same perturbation method as in the Irregular Sampling scenario of (Zeng & Jiang, 2025). The resulting evaluations on the perturbed grid are treated as irregularly sampled data. Three perturbation levels are tested ($\sigma_T \in \{0.1, 0.2, 0.3\}$).We apply a standard FDA preprocessing step: project each curve onto a truncated Fourier basis and reconstruct on a common grid. This preserves low-frequency structure—the most relevant component after SRVF alignment—while ensuring consistent input resolution. Without hyperparameter tuning, our method maintains stable registration and clustering performance across all datasets except **Shapes**, where performance degrades moderately as $\sigma_T$ increases (middle panel, Figure 4).

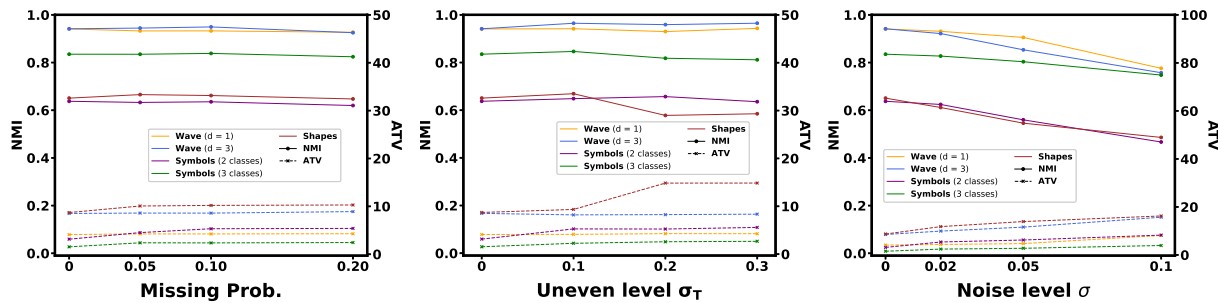

*Figure 4.* Robustness of **NeuralFLoC** to missing data, irregular sampling, and additive noise (left to right).

**Robustness to Noise.** We inject Gaussian noise with $\sigma \in \{0.02, 0.05, 0.10\}$ into Z-score-normalized real-world datasets. As shown in the right panel of Figure 4, performance remains stable across all datasets at $\sigma = 0.02$. For low-to-moderate noise ($\sigma \leq 0.05$), performance degrades gradually; even at $\sigma = 0.10$, the decline is moderate on most datasets. Note that because the raw data already contain intrinsic noise, the resulting noisy observations exhibit considerably higher total variation than the injected noise level alone would suggest.

**Scalability to Large Datasets.** Our method scales efficiently with complexity $O\big(N(CK + CT + T)\big)$ (see Section 3.5). On the augmented 2-class **Symbols** dataset ($\approx 70k$ samples, a $200\times$ scale-up; further scaling was limited by computational resources), the method maintains near-original performance (ATV = 3.2, ACC = 0.942, NMI = 0.683 vs. 3.1, 0.930, 0.639) and outperforms all baselines in both registration and clustering (Table A.3, Appendix C), confirming its advantages persist at scale.

## 7. Conclusion

We introduced **NeuralFLoC**, a fully unsupervised deep-learning framework that jointly addresses functional registration and clustering by integrating Neural ODEs with spectral clustering. By modeling warps as smooth diffeomorphic flows, **NeuralFLoC** aligns curves and learns discriminative templates without supervision, effectively separating phase and amplitude variation. Theoretically, we provide universal approximation guarantees and asymptotic consistency. Empirically, **NeuralFLoC** outperforms state-of-the-art methods in registration (ATV) and clustering (ACC, NMI), while demonstrating robustness to missing data, irregular sampling, noise, and scale.

**NeuralFLoC** has limitations: performance can degrade on trajectories with sharp discontinuities or high-frequency volatility, where smoothness assumptions break, and a joint framework offers little benefit when phase variability is negligible. Future work includes extending **NeuralFLoC** to such challenging trajectories and incorporating uncertainty quantification for the learned warps.

## Acknowledgements

This work was supported by the National Natural Science Foundation of China (NSFC) Youth Science Fund [Grant No. 12301353] and the High-Performance Computing (HPC) platform at ShanghaiTech University.

## Impact Statement

This paper presents work whose goal is to advance the field of Machine Learning. There are many potential societal consequences of our work, none which we feel must be specifically highlighted here.

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

# A. Proof of Theorems 4.1 and 4.2

*Proof of Theorem Theorem 4.1.* We show that arbitrarily small approximation error in the encoder and vector field induces arbitrarily small error in the SRVF registration loss.

**Step 1: Encoder Approximation.** By Assumption (A1) and compactness of the input domain, for any $\delta_1 > 0$ there exists $\Theta_{\text{enc}}$ such that

$$\|\Phi(x_i; \Theta_{\text{enc}}) - z_i^*\|_2 < \delta_1,$$

where $z_i^*$ denotes the latent initial condition generating $\gamma_i^*$.

**Step 2: Vector Field Approximation.** By Assumption (A2), for any $\delta_2 > 0$ there exists $\Theta_{\text{ode}}$ such that

$$\sup_{(\tau, t, z)} \|f^*(\tau, t, z) - f(\tau, t, z; \Theta_{\text{ode}})\| < \delta_2.$$

**Step 3: Stability of ODE Trajectories.** Let $\tau_i^*(t)$ and $\tau_i(t)$ denote the ODE solutions corresponding to $(z_i^*, f^*)$ and $(\hat{z}_i, f)$ respectively. By continuous dependence of ODE solutions on initial conditions and vector fields (via Grönwall's inequality),

$$\|\tau_i^* - \tau_i\|_\infty \le C(\delta_1 + \delta_2),$$

for some constant $C > 0$ depending only on the Lipschitz constant of $f^*$.

Since boundary normalization is a Lipschitz map on strictly monotone trajectories, the induced warping functions satisfy

$$\|\gamma_i^* - \hat{\gamma}_i\|_\infty \le C'(\delta_1 + \delta_2).$$

**Step 4: Continuity of the SRVF Loss.** By Assumption (A4), the SRVF mapping

$$\gamma \mapsto q(x \circ \gamma)$$

is continuous from $(\Gamma, \|\cdot\|_\infty)$ to $L^2([0,1])$ (Srivastava et al., 2011). Hence, there exists $\eta > 0$ such that

$$\|\gamma_i^* - \hat{\gamma}_i\|_\infty < \eta \quad \Rightarrow \quad \|q_i(\gamma_i^*) - q_i(\hat{\gamma}_i)\|_2^2 < \varepsilon/N.$$

Summing over $i = 1, \ldots, N$ yields

$$|R(\gamma^*) - R(\hat{\gamma})| < \varepsilon.$$

$\square$

*Proof of Theorem 4.2.* We establish consistency by analyzing the asymptotic behavior of the joint empirical objective.

**Step 1: Consistency of Registered Functional Representations.** By Theorem 4.1, the learned diffeomorphic warpings converge in probability to the true warpings as $N \to \infty$. Consequently, the registered curves $\tilde{x}_i$ converge uniformly to their population-aligned counterparts. Since the Fourier basis $\{\phi_k\}_{k=1}^K$ is orthonormal in $L^2([0,1])$, the corresponding Fourier coefficients

$$\mathbf{a}_i = \big(\langle \tilde{x}_i, \phi_1 \rangle, \ldots, \langle \tilde{x}_i, \phi_K \rangle\big)$$

converge in probability to the true coefficients $\mathbf{a}_i^*$.

**Step 2: Uniform Convergence of the Clustering Objective.** Let $\mathcal{L}_{\text{clu}}^{(N)}$ and $\mathcal{L}_{\text{clu}}$ denote the empirical and population clustering objectives, respectively. Under Assumption (B2), the Fourier coefficients have finite second moments, and the Student-$t$ kernel defining $p_{ij}$ is smooth and bounded. Hence, by a uniform law of large numbers,

$$\sup_{\{\mu_j\}} \big| \mathcal{L}_{\text{clu}}^{(N)} - \mathcal{L}_{\text{clu}} \big| \xrightarrow{p} 0.$$

**Step 3: Identifiability and Assignment Consistency.** By Assumption (B1), the population clustering risk $\mathcal{L}_{\text{clu}}$ admits a unique minimizer corresponding to the true cluster centroids. Standard results on mixture model estimation imply that the empirical minimizer converges in probability to the population minimizer, yielding

$$\hat{p}_{ij} \xrightarrow{p} p_{ij}^*.$$

**Step 4: Joint Objective Convergence.** The registration loss is continuous with respect to the warping functions and the cluster centroids (Theorem 4.1). Combining Steps 1–3, the empirical joint objective $\mathcal{L}_{\text{total}}$ converges uniformly to its population counterpart. By Assumption (B3), the population minimizer is unique (up to label permutation), hence

$$\mathcal{L}_{\text{total}}^{(N)} \xrightarrow{p} \mathcal{L}_{\text{total}}^*,$$

and the joint registration–clustering estimator is consistent. $\square$

## B. Evaluation Metrics

To evaluate registration quality, we utilize the **Adjusted Total Variance (ATV)**(Srivastava & Klassen, 2016; Jiang et al., 2025), which measures the compactness of aligned classes relative to their separation:

$$\text{ATV} = \frac{1}{\binom{C}{2}} \sum_{1 \leq u < v \leq C} \frac{\text{TV}_{uv}}{d(\text{mean}(\tilde{x}_u), \text{mean}(\tilde{x}_v))}, \tag{A.1}$$

where $\text{TV}_{uv}$ is the intra-class total variation. Lower ATV signifies that the registration successfully reduced phase variability within clusters while maintaining discriminative shape features.

lustering performance is quantified using **Accuracy (ACC)** and **Normalized Mutual Information (NMI)** (Strehl & Ghosh, 2002), defined in Eq. A.2 and A.3:

$$\text{ACC} = \max_m \frac{\sum_{i=1}^n \mathbf{1}\{l_i = m(c_i)\}}{n}, \tag{A.2}$$

where $l_i$ is the ground-truth label, $c_i$ is the predicted cluster, and $m$ iterates over all optimal permutations of labels.

$$\text{NMI}(\Omega, \mathcal{C}) = \frac{I(\Omega; \mathcal{C})}{[H(\Omega) + H(\mathcal{C})]/2}, \tag{A.3}$$

where $I$ is mutual information and $H$ is entropy. Higher ACC and NMI indicate better cluster-class alignment.

## C. More Experimental Results

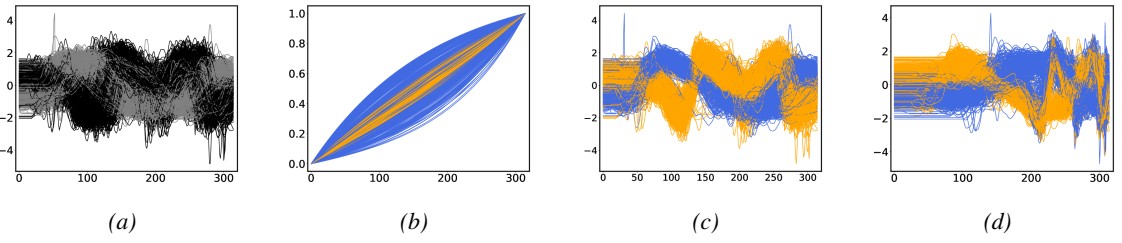

|    (a)    |    (b)    |    (c)    |    (d)    |

*Figure A.5.* (a) Raw data in **Wave** ($d = 1$) dataset with two ground-truth groups indicated by black and grey. (b) Learned warping functions $\gamma(t)$'s by *Ours*. (c) Alignment by *Ours*. (d) Alignment by *Ours (N-ODE → SrvfRegNet)*. Colored curves in (b)-(d) indicate the clustering results.

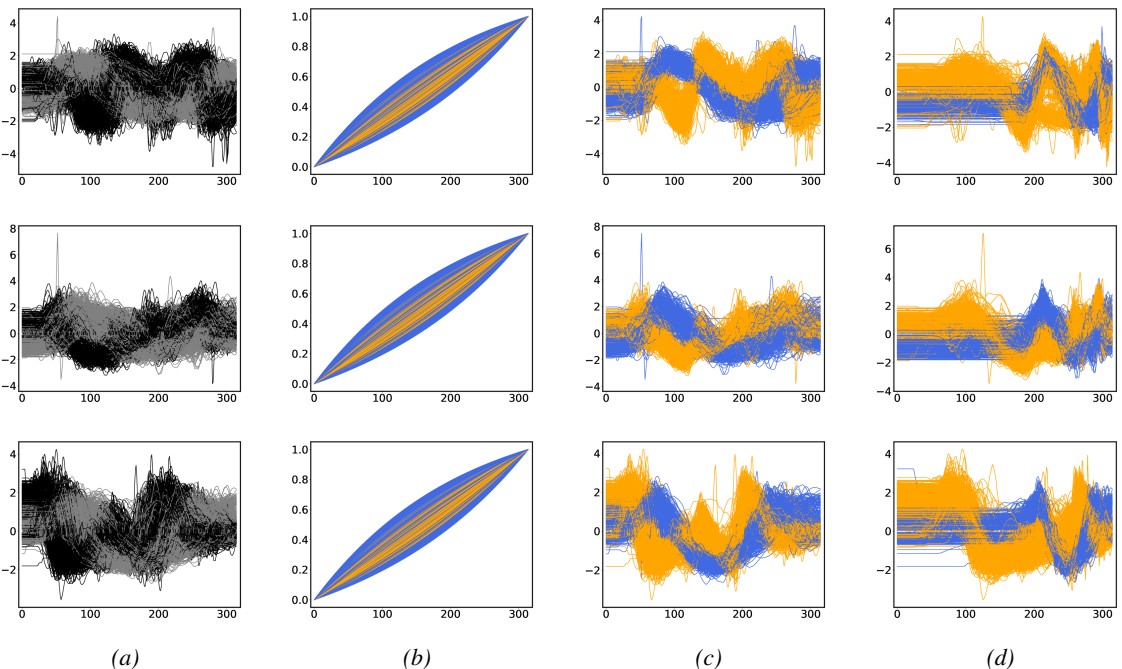

*Figure A.6.* (a) Raw data in **Wave** ($d = 3$) dataset with two ground-truth groups indicated by black and grey. (b) Learned warping functions $\gamma(t)$'s by *Ours*. (c) Alignment by *Ours*. (d) Alignment by *Ours (N-ODE → SrvfRegNet)*. Colored curves in (b)-(d) indicate the clustering results.

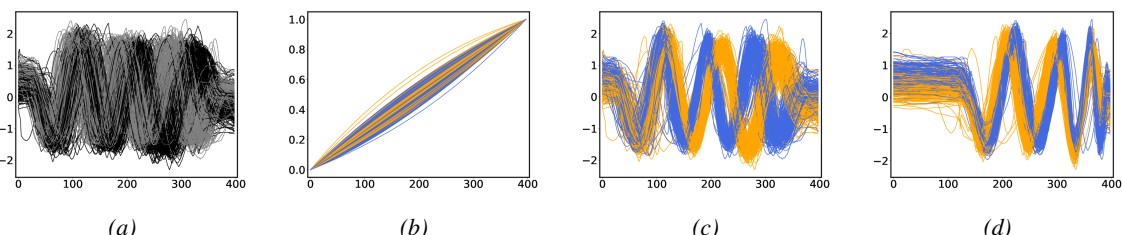

*Figure A.7.* (a) Raw data in **Symbol** (2 classes) dataset with two ground-truth groups indicated by black and grey. (b) Learned warping functions $\gamma(t)$'s by *Ours*. (c) Alignment by *Ours*. (d) Alignment by *Ours (N-ODE → SrvfRegNet)*. Colored curves in (b)-(d) indicate the clustering results.

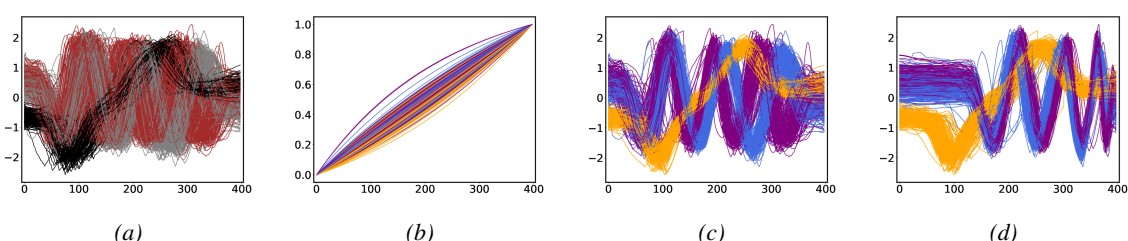

*Figure A.8.* (a) Raw data in **Symbol** (3 classes) dataset with three ground-truth groups indicated by red, black and grey. (b) Learned warping functions $\gamma(t)$'s by *Ours*. (c) Alignment by *Ours*. (d) Alignment by *Ours (N-ODE → SrvfRegNet)*. Colored curves in (b)-(d) indicate the clustering results.

*Table A.2.* P-values from paired t-tests (10 runs) comparing full **NeuralFLoC** against ablated variants. **Bold** indicates significance ($p < 0.05$).

| Hypothesis Tests ($p$-value) | Shapes (2 classes) | | | Wave ($d = 1$) | | | Wave ($d = 3$) | | | Symbols (2 classes) | | | Symbols (3 classes) | | |
|---|---|---|---|---|---|---|---|---|---|---|---|---|---|---|---|
| | ATV | ACC | NMI | ATV | ACC | NMI | ATV | ACC | NMI | ATV | ACC | NMI | ATV | ACC | NMI |
| Full vs. (N-ODE → SrvfRegNet) | **0.0404** | **0.0007** | **0.0015** | **0.0000** | **0.0000** | **0.0000** | **0.0010** | **0.0166** | **0.0043** | 0.3202 | **0.0460** | 0.0543 | 0.1919 | **0.0478** | 0.0758 |
| Full vs. w/o Reg | - | **0.0000** | **0.0000** | - | **0.0000** | **0.0000** | - | **0.0000** | **0.0000** | - | **0.0000** | **0.0000** | - | **0.0000** | **0.0000** |
| Full vs. w/o Clu | **0.0015** | - | - | **0.0039** | - | - | **0.0000** | - | - | **0.0000** | - | - | **0.0000** | - | - |

*Table A.3.* Performance on a large-scale dataset. Results for registration and clustering on the 2-class **Symbols** dataset ($200\times$). **Bold** denotes best performance; the K-Graph baseline fails at this scale.

| Model | ATV | ACC | NMI |
|---|---|---|---|
| SrvfRegNet + FPCA | 7.5 | 0.743 | 0.186 |
| SrvfRegNet + BSpline | 7.5 | 0.648 | 0.062 |
| SrvfRegNet + FAE | 7.5 | 0.502 | 0.000 |
| SrvfRegNet + RandomNet | 7.5 | 0.744 | 0.188 |
| SrvfRegNet + K-Graph | 7.5 | - | - |
| *Ours (N-ODE→SrvfRegNet)* | 3.3 | 0.919 | 0.625 |
| *Ours* | **3.2** | **0.942** | **0.683** |

