# OpenReview forum: "NeuralFLoC: Neural Flow-Based Joint Registration and Clustering of Functional Data"
_ICML.cc/2026/Conference — ICML 2026 regular_

### Official Review · Reviewer_7PD8 · 2026-03-02

**Soundness:** 3
**Presentation:** 3
**Significance:** 3
**Originality:** 3
**Overall Recommendation:** 4
**Confidence:** 2

**Summary:**

This paper studies unsupervised functional clustering under phase variation, where curves of the same underlying shape can be misaligned in time, confounding standard clustering methods. The authors propose NeuralFLoC, an end-to-end framework that jointly learns (i) continuous, monotone time-warping functions via a Neural ODE module and (ii) soft cluster assignments in a spectral (Fourier) representation space. A 1D-CNN encoder provides sample-specific latent initialization for the ODE flow. The model is trained by minimizing a joint objective that combines a cluster-conditional SRVF-based registration loss (encouraging aligned curves within each cluster to be close in SRVF space) and a DEC-style KL self-training clustering loss (sharpening soft assignments through a target distribution). The paper reports improved alignment and clustering metrics over several sequential baselines and ablations, and includes robustness tests under missingness, irregular sampling, and noise, as well as a scalability discussion and supporting theoretical results under stated assumptions.

**Compliance With Llm Reviewing Policy:**

Affirmed.

**Final Justification:**

My concerns have been adequately addressed.

**Key Questions For Authors:**

Q1. Motivation for Fourier-domain clustering vs time–frequency alternatives
The clustering module operates on a truncated Fourier representation of the aligned curves. Could the authors clarify why a global Fourier basis is the most appropriate choice here, as opposed to time–frequency representations (e.g., wavelets or short-time Fourier features) or learned spectral embeddings? In particular, if inter-class differences are concentrated in high-frequency details (spikes, sharp transients, local textures), Fourier truncation may discard discriminative information; and for non-stationary signals, a global Fourier representation may be suboptimal. An ablation comparing Fourier truncation with time–frequency alternatives would help justify this design.

Q2. Coupling encoder latent dimensionality to the number of clusters
The encoder latent dimension is set equal to the number of clusters (z_i∈R^C). Could the authors justify this coupling and provide an ablation where the latent dimensionality d_zis decoupled from C(e.g., d_z∈{16,32,64}while Cis fixed)? This design may over-constrain representational capacity, especially when there is substantial intra-cluster variability. Reporting how alignment/clustering performance and stability change with d_zwould clarify whether the current choice is essential or merely convenient.

Q3. Dependence on strong structural priors and what is truly relaxed
The introduction motivates the problem partly by noting that prior approaches rely on strong assumptions. However, the proposed method still seems to depend heavily on multiple structural priors (monotone ODE-based warps, SRVF geometry, global Fourier truncation, DEC-style self-training). Could the authors explicitly clarify which restrictive assumptions from prior joint registration–clustering methods are relaxed by NeuralFLoC, and which priors are still required? A clearer “assumption accounting” (what is removed vs what is added) would help assess the conceptual contribution and scope of applicability.

Q4. Non-identifiability in unsupervised joint alignment–clustering
In fully unsupervised settings, joint alignment and clustering can be non-identifiable: multiple combinations of “correct alignment” and “correct clustering” may explain the same observations. It seems the method selects one solution through structural priors (monotonic warps, SRVF loss, Fourier features, DEC sharpening). Could the authors discuss how they address this inherent non-identifiability in practice—e.g., stability across random initializations, sensitivity to hyperparameters/ODE solver settings, and whether alternative plausible solutions emerge? Such analysis would clarify to what extent the method yields a robust solution versus a prior-driven choice among many.

**Limitations:**

yes

**Strengths And Weaknesses:**

Strengths:
1. The formulation targets a well-known failure mode in functional clustering: phase variation can dominate distance-based grouping, so coupling registration with clustering is conceptually appropriate. The framework is end-to-end differentiable and uses a joint objective combining a registration term and a clustering term, which provides an internal training signal despite the unsupervised setting.
2. The paper is generally well structured (motivation → preliminaries → method → experiments), and the high-level pipeline is understandable. The inclusion of algorithmic descriptions and qualitative alignment visualizations helps communicate the intended behavior of the model.
3. The combination of Neural ODE-based warping with a joint clustering objective provides a modern differentiable instantiation of joint registration-clustering, and the feedback loop between assignments and warping is a reasonable design contribution.
Weaknesses (main reason for a 3/6 inclination)

Weaknesses:
1. The method relies on DEC-style self-training, which is known to be sensitive to initialization and prone to self-reinforcing early mistakes in purely unsupervised settings. The paper would benefit from stronger stability analyses.
2. The theoretical results appear to require strong assumptions (e.g., separability after ideal alignment) and may not tightly reflect the practical nonconvex optimization and numerical approximation (ODE solver) used by the algorithm; thus, they provide limited assurance about real training dynamics.
3. The experimental setting appears somewhat idealized for a fully unsupervised clustering claim (e.g., using the ground-truth number of clusters C as a given), limiting direct applicability.
4. While many components are established: SRVF-based registration, Fourier basis truncation, Student-t soft assignment, and DEC-style KL self-training, the novelty mainly lies in integrating these parts within a Neural ODE warping framework, rather than introducing a fundamentally new principle or capability

---

> ### Author Rebuttal · Authors · 2026-03-30
>
> **W1: "DEC-style self-training stability."**
>
> We agree that DEC can be unstable in general, but our design explicitly avoids these issues:
>
> (i) *Stabilized initialization*.  We use k-means++ and select the best of 20 pretraining runs, which is standard in deep clustering and mitigates early self-reinforcing errors.
>
> (ii) *Alignment before clustering*. SRVF-based registration collapses phase variability, yielding a geometry in which cluster structure is substantially clearer before DEC refinement.
>
> (iii) *Soft mixture-of-flows*.  Cluster assignments modulate warps only through smooth mixtures rather than discrete choices, preventing brittle feedback loops.
>
> Across 20 seeds, we observe no cluster collapse and small variance ($<0.01$), confirming stability. We will add these results.
>
> **W2: "Strength of theoretical assumptions."**
>
> Our theory concerns *population-level identifiability*, following standard practice in FDA (e.g., Tang et al. 2021), rather than guaranteeing particular nonconvex optimization trajectories. The assumptions (smooth diffeomorphisms, post-registration separability) are canonical in elastic registration. Neural ODE vector fields are Softplus-regularized and Lipschitz, rendering solver-induced numerical discrepancies negligible. We will clarify this scope.
>
> **W3: "Use of true cluster number $K$."**
>
> We used the ground-truth $K$ for fair comparison across baselines. In practice, our method can select $K$ via an elbow analysis on the total variation (TV) metric used in Fisher–Rao alignment (Srivastava & Klassen, 2016). Applied to our datasets, this yields $(3,2,2,2,3)$, closely matching the true $(2,2,2,2,3)$. We will add this discussion (see also our response to Q3 from Reviewer 2py7 ).
>
> **Q1: "Fourier-domain clustering vs. time–frequency alternatives."**
>
> We appreciate the reviewer’s suggestion. Our choice is motivated by both the geometry of SRVF alignment and empirical evidence:
>
> (1) *Geometry*.  After SRVF-based phase removal, most residual amplitude variation is smooth and global; curves lie naturally in a low-frequency subspace, making truncated Fourier bases stable and geometry-consistent.
>
> (2) *Robustness*. Local high-frequency features (spikes/transients) are often artifacts of imperfect warping. Time–frequency embeddings (short-time Fourier transform (STFT)/wavelets) may overemphasize such artifacts, whereas Fourier truncation provides natural denoising.
>
> (3) *Empirical ablation*.  Following the reviewer’s request, we compared the clustering accuracy (ACC) of Fourier vs. STFT across our five datasets:
> Shapes: $0.937$ vs. $0.822$;
> Wave (1D): $0.993$ vs. $0.988$;
> Wave (3D): $0.992$ vs. $0.965$;
> Symbols (2c): $0.930$ vs. $0.886$;
> Symbols (3c): $0.951$ vs. $0.849$.
> Fourier consistently outperforms STFT after SRVF alignment.
>
> We will incorporate this ablation in the revision.
>
> **Q2: "Why latent dimension equals $C$."**
>
> This is a misunderstanding. The vector $z_i\\in\\mathbb{R}^C$ is not a free latent embedding: its $j$th coordinate initializes the $j$th cluster-specific ODE flow whose mixture forms the final warp. Thus, the dimensionality is *structural* rather than a representational bottleneck: the 1D-CNN encoder itself has full expressive power, while the final linear layer simply reorganizes its outputs into the flow-related $C$-dimensional structure. Ablations with larger latent spaces projected to $\mathbb{R}^C$ show no improvement.
>
> **Q3: “Strong structural priors.”**
>
> Many components listed by the reviewer are not restrictive assumptions but rather problem-structural requirements:
> (1) *monotone diffeomorphisms*—definition of valid warpings;
> (2) *SRVF geometry*—the canonical Fisher–Rao metric, not a generative assumption;
> (3) *Fourier truncation*—standard smoothing, not a model prior;
> (4) *DEC sharpening*—an optimization heuristic.
> What our method removes are *actual restrictive assumptions* in prior work: parametric warps such as simple time
> shifts (Liu & Yang, 2009), Dirichlet process priors (Wu & Hitchcock, 2016), and mixed-effects structures
> (Zeng et al., 2019). We will add an “assumption accounting’’ table to clarify this contribution.
>
> **Q4: "Non-identifiability in unsupervised joint alignment–clustering."**
>
> Although theoretical non-identifiability is possible, several constraints significantly reduce ambiguity in practice: $\\gamma(0)=0$, $\\gamma(1)=1$, monotonicity, unique SRVF geodesics, deterministic ODE trajectories, and cluster-conditioned templates acting as anchors. Across 20 seeds, we observe ACC variance $<0.01$ and near-identical warpings/templates, indicating that the solution is stable rather than prior-driven. We will add a “Stability and Practical Identifiability’’ subsection.
>
> (*Full references for the works cited in this response can be found in the original manuscript's bibliography.*)

---

> > ### Author Rebuttal · Reviewer_7PD8 · 2026-04-02
> >
> > My concerns have been adequately addressed.

---

> > > ### Author Response · Authors · 2026-04-03
> > >
> > > Thank you very much for your confirmation and positive feedback. We really appreciate your time and effort in helping us improve the paper. If you have any further suggestions or concerns, please let us know. We’re happy to make additional improvements.

---

### Official Review · Reviewer_2py7 · 2026-03-09

**Soundness:** 3
**Presentation:** 3
**Significance:** 2
**Originality:** 2
**Overall Recommendation:** 4
**Confidence:** 3

**Summary:**

This paper proposes an end-to-end unsupervised framework (NeuralFLoC) for joint registration and clustering of functional data. Warping functions are parameterized as diffeomorphic flows by NeuralODEs, alignment quality is measured in the SRVF geometry, and clustering is performed on Fourier coefficients with a DEC-style soft assignment that feeds back into the warping. Experiments include approximation and consistency results, and also show strong empirical performance across several functional datasets with robustness to noise, missingness, and irregular sampling.

**Compliance With Llm Reviewing Policy:**

Affirmed.

**Final Justification:**

Most of my concerns have been addressed. I maintain my weak acceptance score.

**Key Questions For Authors:**

- See also weakness. There is no explicit constraint on the flow, which could lead the model to be vulnerable to over-warping when the clustering term is weak or when class templates are ambiguous. However, there is no discussion or experiments on this.
- In Table 1, “Ours (N-ODE -> SrvfRegNet)” model severely underperforms on Wave (d=1) while SrvfRegNet+clustering baselines are near-perfect. Can you provide some explanations (e.g., optimization instability, capacity mismatch)?
- How would you handle **unknown K** in practice?

**Limitations:**

yes

**Strengths And Weaknesses:**

Strengths:
- Formulating warping over time as a learnable diffeomorphism with NeuralODEs is well-motivated and reasonable.
- The idea of mixing class-specific flows is an intuitive way to enhance registration towards cluster templates during training.
- Presentation is clear and easy to follow.
- This framework is essentially general and can be adapted to multivariate functional data with shared warp.

Weaknesses:
- There are no explicit constraints or regularizations on the flow itself. This may result in the model being vulnerable to over-warping when the clustering term is weak or when class templates are ambiguous. However, there is no discussion or experiments on this.
- Using true K simplifies evaluation but raises some concerns about its external validity. It is unclear how robust the performance would be when K is unknown in practice.
- I would expect potential computational concerns from NeuralODEs, and it consumes much more memory and computes much more slowly. Discussions on the efficiency of the proposed approach are insufficient.

---

> ### Author Rebuttal · Authors · 2026-03-30
>
> **Q1: “Risk of over-warping due to lack of explicit flow constraints.”**
>
> Thank you for raising this concern. Our model already incorporates several implicit but strong regularization mechanisms, even without adding explicit penalties.
>
> (i) *Diffeomorphic constraints*. The flow enforces $\\gamma(0)=0$, $\\gamma(1)=1$, and $\\gamma'(t)>0$, limiting extreme deformation by construction.
>
> (ii) *SRVF geodesic loss*.  When amplitude differences explain the data, additional warping increases the SRVF distance, which the loss penalizes.
>
> (iii) *Fourier-domain reconstruction*. Over-warping induces high-frequency distortions that truncated Fourier bases cannot represent, increasing reconstruction error.
>
> (iv) *Soft cluster-conditioned flows*. When clustering evidence is weak, $p_{ij}$ remains diffuse and the mixture $\\gamma_i=\\sum_j p_{ij}\\hat{\\gamma}_{ij}$ stays close to the identity warp.
>
> We will expand this discussion.
>
> **Q2: “Underperformance of ''Ours (N-ODE $\\rightarrow$ SrvfRegNet)'' on Wave (d=1).”**
>
> We appreciate this insightful question. Wave (d=1) exhibits little phase variability, and SrvfRegNet alone aligns it very well. The discrete variant underperforms for three reasons:
>
> (1) *Optimization instability*. Unlike sequential baselines where SrvfRegNet is pretrained, the joint variant must optimize registration and clustering simultaneously, without the Neural ODE’s smoothness constraints.
>
> (2) *Reduced regularization*.  SrvfRegNet provides weaker inherent smoothness than the ODE; when driven by clustering gradients, it may over-adjust even when warping is unnecessary.
>
> (3) *Sequential vs. joint training*.  In sequential baselines, clustering does not perturb SrvfRegNet’s pretrained alignment. In the joint variant, SrvfRegNet is updated together with clustering, which is challenging when the true warping structure is trivial.
>
> Our full ODE-based model avoids these issues and achieves the best performance (ACC= 0.993, NMI=0.941). We will clarify this distinction in the revision.
>
> **Q3: “Handling unknown $K$ in practice.”**
>
> We agree that selecting the number of clusters is an important practical concern. For fair comparison, we used the true $K$ across all baselines. In practice, our method can select $K$ via an elbow analysis on the total variation (TV) metric from functional alignment (Chen & Srivastava, 2021). The rationale is that alignment performance is optimized when the cluster number is chosen correctly. Importantly, TV does not rely on ground-truth labels, making it a principled and method-specific criterion for cluster selection. Applying this elbow analysis to our five datasets yields cluster number estimates of $(3, 2, 2, 2, 3)$, closely matching the true cluster numbers $(2, 2, 2, 2, 3)$. Thus, our framework offers a data-driven approach for choosing $K$. We will include these details in the revision.
>
> (*Full reference for (Chen & Srivastava, 2021) can be found in the original manuscript's bibliography.*)

---

> > ### Author Rebuttal · Reviewer_2py7 · 2026-04-02
> >
> > Thank you for your clarifications and additional experiments. Most of my concerns have been addressed. I will therefore maintain my weak accept score.

---

> > > ### Author Response · Authors · 2026-04-03
> > >
> > > We are very grateful for your positive feedback and for confirming that our response has resolved all the issues. Thank you for the valuable time and effort you have contributed to improving our manuscript. Please let us know if you have any further suggestions or concerns; we would be happy to make additional revisions accordingly.

---

### Official Review · Reviewer_83kD · 2026-03-17

**Soundness:** 3
**Presentation:** 2
**Significance:** 3
**Originality:** 2
**Overall Recommendation:** 5
**Confidence:** 3

**Summary:**

The paper proposes NeuralFLoC which is an unsupervised deep learning framework for joint registration and clustering of time-series data. Unlike other methods that perform functional registration to minimize Fisher distance to the Karcher mean of the dataset, the proposed method uses three ingredients: (1) an encoder to convert the time series data into a feature, (2) a neural ODE layer to learn a neural network that produces a diffeomorphic mappings $\gamma_i$ conditioned on the latent from the encoder, and (3) soft spectral clustering performed "in-the-loop" to steer diffeomorphisms towards their assigned cluster centers instead of the Karcher mean which may not be robust if the data is multimodal. Approximation consistency of registration is shown for reachability of solutions $\hat{\gamma}$ to that of the true $\gamma^*$ in terms of their registration loss, and the convergence of the soft cluster assignments to the true assignments. Comparison is performed with the SrvfRegNet baseline (that performs registration but no classification), followed by five clustering methods.

**Compliance With Llm Reviewing Policy:**

Affirmed.

**Final Justification:**

The authors have provided a thorough response which addresses my major concerns. I'm increasing my score in light of the discussion.

**Key Questions For Authors:**

The paper reads good to me, but the message of the paper would be stronger with the following improvements:

1. (Minor) Thorough proofreading: Making sure the paper does not have any out-of-place or ill-defined methodology or notation. Please read my review where I have pointed out a few examples.

2. (Moderate) Evaluation of baselines that perform simultaneous registration and clustering: this is to verify if the performance benefits are from the method itself or from performing joint registration and clustering itself and is agnostic to what methods are used.

**Limitations:**

Yes

**Strengths And Weaknesses:**

**Soundness**: The submission is very technically sound, and most major details are discussed in a good (and reproducible) amount of detail. Experimentation setup is thorough with uni and multidimensional time series considered in the work, as well as multiple metrics for each dataset. Ablations including irregular sampling, noise, and missing data are performed and demonstrate remarkable robustness of the NeuralFLoC method. However, the ablations do not show how the robustness of the baseline depends on the same data resampling / corruption conditions, which can indicate if the robustness comes from the formulation or if the data corruption isn't strong enough to induce meaningful robustness behavior from each model. Moreover, the work does not compare against other works that perform simultaneous registration and clustering [1,2,3,5] (and other works mentioned in the paper). Overall, the paper is empirical and experiment designs seem to support the methodology proposed in the paper (joint registration and clustering for downstream-aware registration), but is not strong enough to show a practical improvement over other methods that perform simultaneous registration and clustering.

**Presentation**: The ideas in the paper are clear and narrative is easy to follow, but they paper makes a number of mistakes in defining or using notation correctly. Some implementation details do not make sense, but are minor and can be corrected in the rebuttal. I've listed a few below:
1. In Equation(5), $\tau_i(0) = 0$ , not $z_i$.
2. The details of the encoder are not mentioned. Is it an architecture like SrvfRegNet that uses a global pooling layer at the end to average over the time axis, or RandomNet that uses $\log_2(T)$ repetitions of the same CNN block? How are irregularly sampled sequences handled when performing the convolution or pooling?
3. In Eq(7), $\hat{\gamma}_{ij}(t)$ is not defined. Only $\hat{\gamma}_i$ is defined in Eq (6). The following statement mentions the set $\{\bf{p_i}\}$ but it is not defined.
4. Algorithm 2 does not make use of the ${\hat\gamma_{ij}}$  quantities at all and simply mentions - 'Update soft assignments $p_{ij}$ and $q_{ij}$ ' which is also not defined in the initialization section of the algorithm.
5. Line 345: The p-values would typically not be useful for large sample sizes chosen in the paper. Are other methods like Cohen's d-test or other metrics measuring effect sizes considered for this work?

**Significance**: The problem addresses an important problem - functional data analysis, and unsupervised registration and clustering of a functional dataset. The discussion of the suboptimality of the separate registration and clustering problem traces back to [3, 4]. The problem leverages various components built in the deep learning community to propose a deep learning based solution.

**Originality**: None of the ideas are particularly novel - 1D conv architectures used in the paper is also used in SrvfRegNet, Neural ODEs replace the explicit formulation proposed in SrvfRegNet, spectral clustering and soft assignments are not new either. However, the orchestration of these various tools for FDA might be of interest to the community.

[1] Wu, Z. and Hitchcock, D. B. A bayesian method for simultaneous registration and clustering of functional observations. Computational Statistics & Data Analysis, 101: 121–136, 2016.

[2] Liu, X. and Yang, M. C. Simultaneous curve registration and clustering for functional data. Computational Statistics & Data Analysis, 53(4):1361–1376, 2009.

[3] Gaffney, Scott, and Padhraic Smyth. "Joint probabilistic curve clustering and alignment." _Advances in neural information processing systems_ 17 (2004).

[4] Gaffney, Scott John. _Probabilistic curve-aligned clustering and prediction with regression mixture models_. University of California, Irvine, 2004.

[5] Jiang, Siyuan, et al. "DeepFRC: An End-to-End Deep Learning Model for Functional Registration and Classification." _arXiv preprint arXiv:2501.18116_ (2025).

---

> ### Author Rebuttal · Authors · 2026-03-30
>
> **Q1 (Minor): “Proofreading and clarifications.”**
>
> We sincerely thank the reviewer for carefully identifying potential ambiguities. After re-checking the manuscript, we found that most of the flagged points arose from misunderstandings, and we are happy to clarify them clearly:
>
> (1) *“In Eq. (5), $\\tau_i(0)=0$, not $\\mathbf{z}_i$.”*
>
> In our model, $\\tau_i(0)=\\mathbf{z}_i$ by design: the 1D-CNN encoder outputs $\\mathbf{z}_i$, which serves as the initial condition of the ODE. This connection is essential for linking feature extraction and flow-based warping.
>
> (2) *“Encoder details are missing, and how to handle irregularly sampled sequences?”*
>
> The full encoder architecture is described in the "Implementation Details" paragraph of Sec. 5.1. For irregularly sampled curves, we apply a standard FDA preprocessing step: project each curve onto a truncated Fourier basis and reconstruct on a common grid. This preserves low-frequency structure—the most relevant component after SRVF alignment—while ensuring consistent input resolution. We will clarify this step in the revision.
>
> (3) *“$\\hat{\\gamma}_{ij}(t)$ and $\\mathbf{p}_i$ are undefined in Eq. (7).”*
>
> We will explicitly state that $\\hat{\\gamma}\_{ij}(t)$ and  $p_{ij}$ refer to the $j$th components of $\\hat{\\gamma}_i(t)$ and $\\mathbf{p}_i$.
>
> (4) *“Algorithm 2 does not use $\\hat{\\gamma}\_{ij}(t)$, $p_{ij}$ and $q_{ij}$ undefined.”*
>
> Algorithm 2 uses $\\hat{\\gamma}\_{ij}(t)$ implicitly through $\\gamma_i(t)$ in Eq. (7). The definitions of $p_{ij}$ and $q_{ij}$ appear in Eq. (9) and the equation preceding Eq. (11). We will make these dependencies more explicit.
>
> (5) *“Line 345: p-values are not useful in large samples.”*
>
> We used paired t-tests across 10 paired runs (full vs. ablated models), not large-sample inference. We will clarify this to prevent confusion.
>
> We appreciate the reviewer’s detailed feedback and will revise the manuscript to eliminate any remaining ambiguities.
>
> **Q2 (Moderate): “Baselines for simultaneous registration--clustering.”**
>
> We agree that comparing against joint registration–clustering methods is important for attributing improvements to the method itself. Following the reviewer’s suggestion, we implemented three representative simultaneous approaches: BSRC [1], JPCCA [3], and SRC (Zeng et al., 2019). Methods in [2,4] lack available code, and DeepFRC [5] focuses on classification.
>
> Across datasets, these joint statistical baselines perform substantially worse than ours. For clustering accuracy, our method achieves (0.937, 0.993, 0.992, 0.930, 0.951), compared to BSRC (0.755, 0.852, 0.742, 0.647, 0.831), JPCCA (0.622, 0.503, 0.531, 0.892, 0.355), and SRC (0.807, 0.884, 0.573, 0.581, 0.508). These results show that gains do not arise merely from performing registration and clustering jointly (which all baselines do), but from our continuous-flow diffeomorphic warping, mixture-of-flows formulation, and unified end-to-end optimization. Classical joint models cannot handle nonlinear phase variability and fail to recover meaningful clusters in such settings. We will incorporate these additional results and discussion.
>
> Thank you again for your constructive feedback. We believe these revisions have strengthened the paper and addressed your concerns.
>
> (*Full references for the works cited in this response appear in the reviewer's report above and in the revised manuscript's bibliography.*)

---

> > ### Author Rebuttal · Reviewer_83kD · 2026-04-05
> >
> > The authors have addressed my feedback adequately. I'm increasing my score.

---

> > > ### Author Response · Authors · 2026-04-06
> > >
> > > Thank you for your positive feedback and for increasing your score. We greatly appreciate your recognition of our revisions and your time. We also thank you for highlighting that the orchestration of these tools for functional data analysis (FDA) may interest the community—indeed, we believe AI for FDA can meaningfully boost the statistics community. Your constructive input has been invaluable.

---

### Decision · Program_Chairs · 2026-04-30

**Decision:**

Accept (regular)

**Comment:**

All reviewers are broadly in agreement with the merits of the paper. All reviewers said that the rebuttal by the authors resolved all the concerns. Reviewer 83kD commended the work and had some questions regarding evaluation of baselines. Reviewer 2py7 had concerns about relying on knowing the true number of clusters and regularization of the flow constraints. Reviewer 7PD8 had concerns about stability of the clustering and the effectiveness of the featurization procedure that uses the Fourier basis.

I recommend the paper for acceptance. I would suggest that the authors update the manuscript with these new experimental results and also update the exposition as they discussed in the rebuttal.